# 3D Back Contour Metrics in Predicting Idiopathic Scoliosis Progression: Retrospective Cohort Analysis, Case Series Report and Proof of Concept

**DOI:** 10.3390/children11020159

**Published:** 2024-01-26

**Authors:** Milan Patel, Xue-Cheng Liu, Kai Yang, Channing Tassone, Benjamin Escott, John Thometz

**Affiliations:** 1Department of Orthopedic Surgery, Children’s Wisconsin, Medical College of Wisconsin, Greenfield, WI 53227, USA; 2Division of Biostatistics, Institute for Health and Equity, Medical College of Wisconsin, Milwaukee, WI 53226, USA

**Keywords:** adolescent idiopathic scoliosis, radiography, surface topography, predictive modeling

## Abstract

Adolescent Idiopathic Scoliosis is a 3D spinal deformity commonly characterized by serial radiographs. Patients with AIS may have increased average radiation exposure compared to unaffected patients and thus may be implicated with a modest increase in cancer risk. To minimize lifetime radiation exposure, alternative imaging modalities such as surface topography are being explored. Surface topography (ST) uses a camera to map anatomic landmarks of the spine and contours of the back to create software-generated spine models. ST has previously shown good correlation to radiographic measures. In this study, we sought to use ST in the creation of a risk stratification model. A total of 38 patients met the inclusion criteria for curve progression prediction. Scoliotic curves were classified as progressing, stabilized, or improving, and a predictive model was created using the proportional odds logistic modeling. The results showed that surface topography was able to moderately appraise scoliosis curvatures when compared to radiographs. The predictive model, using demographic and surface topography measurements, was able to account for 86.9% of the variability in the future Cobb angle. Additionally, attempts at classification of curve progression, stabilization, or improvement were accurately predicted 27/38 times, 71%. These results provide a basis for the creation of a clinical tool in the tracking and prediction of scoliosis progression in order to reduce the number of X-rays required.

## 1. Introduction

Idiopathic Scoliosis (IS) is a spinal abnormality commonly defined as a Cobb angle ≥ 10 degrees on anterior posterior radiograph (AP). Adolescent Idiopathic Scoliosis (AIS) has the greatest prevalence, making up 80–85% of total idiopathic scoliosis cases with population prevalence ranging from 0.47 to 5.2% and a 1:1.5 to 3:1 female to male ratio [1,2]. 

Radiography has been the gold standard in the evaluation and progression monitoring for AIS. This practice comes with two potential downsides. The first is that spinal deformities are often multiplanar and require 3D evaluation for adequate correction though AP radiography alone. Secondly, serial radiographs expose children to increased radiation during adolescence, hypothetically increasing cancer risk [3,4,5]. Clinical practice based on the 2007 SOSORT consensus notes that convention in monitoring scoliosis should entail initial AP and lateral radiograph and that patients with stable curvatures requiring no intervention should be imaged no more than once per year with only one AP view [6]. Curve progression is a major concern in those undergoing a growth spurt during puberty, and once skeletal maturity has been reached, more invasive treatments are often necessary to correct IS concerns [7]. These factors necessitate methods with reduced radiation exposure while still facilitating multidimensional evaluation. EOS radiography, which has an estimated cumulative dose reduction of 50.6%, allows us to reconstruct the 3D spine though simultaneous AP/Lateral radiographs in a reasonable time with high re-liability using the sterEOS software (Version#1.5.4.8189) [8,9,10]. SOSORT even suggested spinal radiographic intervals for patients ranging from 6 months to 24 months based on age/Risser staging for skeletal maturity [11]. Additionally, there has been increasing emphasis on improving digital health evaluation through the use of artificial intelligence (AI). Recent studies employ AI in the evaluation of scoliosis and prediction through machine-based learning [12]. These evaluations are still based on the use of radiographic measurements. Although reducing lifetime radiation dosages is a step in the right direction, the elimination of repeat radiation exposure with adequate evaluation of the spine should be the ultimate goal of AIS treatment. The gold standard is to provide a closed-loop system where the patients can have real-time interaction with their curve progression and evaluation while reducing the potential harm of repeat radiographic exposure. 

Surface topography (ST) is an alternative imaging method that tackles the two previously mentioned concerns. The DIERS Formetric 4D system (DIERS International GmbH, Schlangenbad, Germany) enables automatic, dynamic spine mapping through anatomical landmarks of the spine and pelvis for measurement by video rastersterography. This program allows for evaluation of a host of parameters of the spine to be measured by camera to give an estimated 3D reconstruction of the spine. This method of measurement shows strong test–retest reproducibility, intraobserver reliability, and interobserver reliability as an alternative modality in the evaluation of the spine through back contour [13,14]. Previous studies have shown surface topography to be a promising alternative, though evaluation using these methods has only gained popularity since the early 2000s [15,16]. However, ST faces its own challenges such as changes in back contour biasing data such as BMI and uneven muscular development altering overlying topography [17,18]. These alterations can result in erroneous representations of the underlying spine and require some consideration to ensure adequate reconstruction. Despite being a relatively new digital evaluation method, ST allows for real-time, software based, contact-less evaluation of scoliotic curvatures and gait analysis.

Statistical models have been widely used to predict the AIS curve progression using certain patient-specific and radiological risk factors. Tan et al. found that gender, pubertal status, initial Cobb angle and age were important predictors of increasing Cobb angle in the long term [19]. This study suggested a logistic model which predicted if AIS curves would deteriorate beyond 30 degrees. Nault et al. formulated a general linear model evaluating first-visit information to reconstruct the Cobb angle [20]. By using the backward variable selection, their model contained five significant predictors: skeletal maturity, curve classification, Cobb angle, the plane of maximal curvature, and 3D disk wedging (T3–T4, T8–T9). Recently, using machine learning methods and the random forest model, there have been attempts to evaluate curve dynamics over time [21,22]. However, all the methods mentioned above make the prediction based mainly on the radiological variables. Currently, there does not appear to be existing literature that uses non-radiographic factors such as axial surface rotation (ASR) for the prediction of curve progression.

The objective of this study was to create a stratified predictive model which could recreate Cobb angles using non-radiographic means and predict future scoliotic curve progression into curves that will progress, stabilize, or improve.

## 2. Materials and Methods

Patients between 8 and 18 years of age, with IS or spinal asymmetry, having received radiographic evaluation via an EOS Image system (EOS imaging, Paris, France) or plain film radiographs (PACS via Epic) and surface topographic evaluation via a DIERS Formetric 4D system were included in the retrospective study. Patients were seen at the Orthopedic Clinic or Musculoskeletal Functional Assessment Center at Children’s Wisconsin between the dates of 15 August 2018 and 3 January 2022 based on available clinic data. This study was approved by Children’s Wisconsin IRB. Scoliosis Research Society (SRS) criteria for clinical evaluation stratify patients into three categories based on the change between initial visit and follow-up: Improving: ≥6°; Stable −5° to 5°; Progression ≤ −6° [23]. 

Radiographic imaging was collected using traditional AP/Lateral Radiographs or the EOS imaging system reconstructed with the sterEOS software. Radiographs via EOS or non-EOS were evaluated and reported in Epic by Orthopedic Surgeons or Radiologists at Children’s Wisconsin and the Medical College of Wisconsin. Surface topography measurements were made using a DIERS Formetric 4D system (DIERS International GmBH, Schlangenbad, Germany) and its corresponding software. In the examination room, the subject took off clothing on their upper half and put on an apron that covered the entire frontal trunk area. Once the subject was comfortable with their placement of the apron, five markers were placed on the spinal process of the first thoracic vertebrae, posterior superior iliac crests, and shoulders; then, the subject was asked to step up onto the treadmill with their heels lined up with the black line in the center of the treadmill. Meanwhile, the camera length was adjusted to the correct height for the subject, and finally, the scans were taken with 13 total frames in 6 s using light grid projection. The back contour was ascertained by the system. The DIERS Formetric 4D system allows for the measurement of 85 formetric parameters of which 12 metrics are usually reported in clinical examination. These include 3D measurements of trunk length, scoliotic angles, sagittal imbalance, coronal imbalance, pelvic obliquity, pelvic torsion, pelvic inclination, kyphotic angle, lordotic angle, maximum/minimum ASR, and apical ASR. These measurements then allow the software to estimate the curvature of the spine. Imaging modalities were compared within 6 months to minimize curve differences.

Statistical analyses were performed using R version 4.1.3. A *p*-value of less than 0.05 was considered statistically significant. Different demographic and surface topographic factors such as age, gender, ASR from T1 to L5, scoliotic angle, and pelvis surface rotation were considered in building a linear mixed-effects model for predicting the Cobb angle. A proportional odds logistic model was performed to study the association between the covariates mentioned above and the ordinal outcome about the category of future scoliotic curve progression. Backward model selection based on the Akaike Information Criterion (AIC) was used to select the important covariates and determine the best predictive models for the Cobb angle and the scoliotic curve progression, respectively [24]. Unless stated otherwise, all statistical tests were two-sided.

## 3. Results

### 3.1. Demographics

Thirty-eight children who met the inclusion criteria were recruited for the study. The mean age was 13.82 years (SD = 1.61) with 71.1% (n = 27) females and 28.9% (n = 11) males. According to the SRS criteria, eight patients were found to have progressing curves, 19 patients were found to have stable curves, and 11 patients were found to have improved curves. The average follow up was 34.87 months (SD = 24.72) for radiographs and 15.47 months (SD = 16.80) for ST.

### 3.2. Predictive Modeling

Using backward model selection targeting the Cobb angle, the variables of age, gender, ASR at T6, T7, T8, and scoliotic angle were included in the final model as the best predictors. Based on this,
(1)Cobb angle=26.243−1.137Age−7.689Gender−3.590T6+5.912T7−2.684T8+0.729Scoliotic angle
were determined to be the best predictors of Cobb angle using non-radiographic means (Table 1). Gender = 0 for female and 1 for male. The R-squared of the predictive model is 0.869 (*p* < 0.05). 

After AIC variable selection, the measure incorporated six predictive factors during the initial visit encompassing age ASR at T8-10 and ASR at L3-4 in the risk stratification probabilities.
(2)$$\log\left(\frac{\mathrm{Pr}\left(Progression\right)}{1-\mathrm{Pr}\left(Progression\right)}\right)=-12.00+0.75Age-1.25T8+2.77T9-1.74T10+0.72L3-1.19L4$$
(3)$$log(\frac{\mathrm{Pr}\left(Progression\;or\;Stable\right)}{1-\mathrm{Pr}\left(Progression\;or\;Stable\right)})=-8.35+0.75Age-1.25T8+2.77T9-1.74T10+0.72L3-1.19L4$$

The probability of belonging to the progressing group is denoted as Pr (progression), while Pr (progression or stable) represents the probability of belonging to either the progression or stable group. By simple calculation, the predicted probabilities from the model are given below.

Progression:(4)$$Y_{1}=\frac{exp(-12.00+0.75Age-1.25T8+2.77T9-1.74T10+0.72L3-1.19L4)}{1+exp(-12.00+0.75Age-1.25T8+2.77T9-1.74T10+0.72L3-1.19L4)}$$

Improving:(5)$$Y_{3}=\frac{1}{1+exp(-8.35+0.75Age-1.25T8+2.77T9-1.74T10+0.72L3-1.19L4)}$$

Stable:
Y_2_ = 1 − Y_1_ − Y_3_(6)

$Y_{1}$, $Y_{2}$ and $Y_{3}$ denote the predicted probabilities for the progression, stable, and improving groups, respectively. Patients are assigned to the group with the highest predicted probability value among these three based on their individual probabilities (Table 2). 

From Table 2, it is evident that both the lower and upper limits of the range for Y_1_ are the largest value when compared to those for Y_2_ and Y_3_ in the progression category. Consequently, when the true category is progression, Y_1_ has the highest likelihood of being the largest among Y_1_–Y_3_. In other words, in most scenarios, the predicted category should be the same as the true category progression. We have similar conclusions for stable and improvement categories. Overall, the stratified predicative model allowed for 27/38 or 71% of patients to be accurately categorized into their prospective scoliotic curve classification groups (Table 3). The accuracy rates of classification were different among three category groups, and they were 85%, 75%, and 45% for the stable group, the progression group and the improving group, respectively (Table 3).

### 3.3. Case Report and Proof of Concept

To test our modeling, we randomly selected a patient from our study (38 subjects) to evaluate. A 14-year-old girl diagnosed with AIS was chosen. Over a period of approximately 3 months of imaging follow-up, her surface topographic measures evolved. Initially, the scoliotic angle value was 16° with the major curvature positioned between T9 and L3. In the subsequent assessment, the scoliotic measure increased to 26°, and the major curvature was detected between T10 and L4 in the PA view of surface topography (Figure 1).

During a 6-month period, the PA X-rays of the same patient revealed a progression in the Cobb angle from 23° during the initial clinical visit to 33° on follow-up. Both radiographic measurements consistently indicated the major curvature centering around T10-L3, as depicted in Figure 2.

While examining axial plane trends at each spinal level through ASR analysis for the patient, a notable enhancement in the maximal ASR was observed. Specifically, there was an increase in counterclockwise rotation from 8° to 13° at the L2 level for the left major curve, which was evident during the comparison of the two clinical visits (Table 4, Figure 3). In contrast, there was a reduction in clockwise rotation from 13° to 9° at the T8 for the minor curve between T1/T2 and T9 (Table 4, Figure 3). 

While using the stratified predicting model calculated the value of probability at the beginning of the clinical visit, the curve could be classified into Y_1_ as a progression group, since Y_1_ is the largest (Table 5).

## 4. Discussion

This study seeks to demonstrate a proof of concept for the development of a risk stratification and ST-based predictive framework for assessing AIS. Our model suggests that individuals of a younger age and female gender are indicative of curve progression, although the role of gender is a subject of debate [25]. Although our study did not specify an exact age at risk for curve progression, another study considered it as one of the patient-specific risk factors, but it was not a significant risk factor if the patient was diagnosed with AIS at less than 13 years old [26]. However, their systematic review was not in favor of consideration of gender as a significant risk factor [26]. Backwards model selection allowed for the best linear mixed models to be chosen for the prediction of Cobb angle in the future. This model was determined to have an r-squared of 0.869, which means that 86.9% of the variability in the target statistic of future Cobb angle can be explained by non-radiographic surface topographic and demographic measures. Furthermore, when attempting to categorize spinal progression, the model successfully identified 27 out of 38 patients, achieving a classification accuracy of 71%. However, this modeling comes with limitations in that the predictive performance is based on the original performance of the ST-based predictive model for Cobb angle. While the overall accuracy of the stratified predictive model is 71%, it accurately predicted only 45% of the improvement group with the remaining classified as stable. Consequently, the model performs better for stable and progressing curves but shows a weaker performance in predicting improving curves, indicating potential for improvement.

Alfraihat et al. developed a random forest model to find the most important prognostic markers for progression of the curve angle and to predict Cobb angle [21]. The final random forest model is complex with 15 predictors: initial major Cobb angle, initial lumbar lordosis angle, initial thoracic kyphosis angle, age at initial follow-up, age at final follow-up, initial apical wedge angle, the time span between age at initial and final follow-ups, flexibility, axial rotation, Lenke type, gender, brace status, apex location, number of levels involved in the curve, and Risser sign. Based on a deep convolutional neural network, Yahara et al. used the X-ray images of three regions of interest including lung, abdomen, and total spine to predict the risk of scoliosis progression in patients with AIS [22]. Although the machine learning techniques are powerful in some applications, they are “black box” models, which are difficult to interpret and justify. In a systematic review evaluating curve progression, the top five factors that influence or predict curve progression include initial curve magnitude, skeletal maturity, curve location, age, and status of menarche [26], of which the most significant predictive risk factors were curve magnitude (Cobb angle > 25°), skeletal maturity (either Risser stage < 1 or Sanders Maturity Scale < 5), and curve location (single or double thoracic). Our ST-based predictive model shared several common variables, such as the initial major Scoliotic angle, thoracic curve location from T6 to T8 or T8 to T10, and age at initial follow-up, although our model was derived from the outlined statistical methods and non-radiographic measurements.

The outcome regarding the category of future scoliotic curve progression is a categorical variable with three levels: progression, stable and improving. Thus, the classification methods such as the proportional odds logistic model, linear (or quadratic) discriminant analysis, support vector machine, random forest or other machine learning approaches for classification can be employed to predict the outcome. It is important to note that there is a natural order among the three categories: progression, stable and improvement. We choose the proportional odds logistic model because it is the only classification method that can leverage such ordinal information in model fitting. Furthermore, the proportional odds logistic model has another two major advantages compared to alternative methods. First, this model is relatively simple with fewer parameters than other methods for the multiclass classification problem. When faced with two models exhibiting similar performance, it is almost always preferable to choose the simpler model. Second, the proportional odds logistic model provides highly interpretable coefficients that quantify the relationship between the covariates and the outcome variable. These coefficients often come with confidence intervals and *p*-values, further enhancing the model’s interpretability. In contrast, many machine learning methods, such as random forest and neural network, are black boxes, making them challenging and even infeasible to interpret. Moreover, *p*-values, widely used for evaluating statistical significance, cannot be obtained from these complex machine learning methods unless resampling-based methods such as bootstrap and permutation are used. However, these resampling-based methods are computationally intensive, and a very large sample size is required to ensure that they can estimate *p*-values accurately. Thus, the proportional odds logistic model is used to predict future scoliotic curve progression when performing data analysis [27]. With respect to the risk factors in the final model, they are selected using the AIC criteria, which is capable of choosing the predictive model with the best predictive accuracy [23]. 

The proportional odds logistic model is mainly developed to characterize the linear relationship between the covariates and the log odds ratio of the outcome variable. When the relationship is not linear, random forest and neural network models are expected to demonstrate superior predictive performance. 

There has been increasing emphasis on the development of non-ionizing, radiation-free evaluation methods for the progression of AIS. While there have been promising improvements in the use of surface topography, there is still uncertainty in its viability for clinical use, as it is a proxy rather than a direct representation of the spine [28,29]. While radiographs still reign as the gold standard imaging of choice during initial evaluation and during critical periods of development, ST has shown utility in the reduction in use of number of radiographs in a long-term of clinical follow-up of curve progression. Since the COVID-19 pandemic, the WHO has put forward guidance in the topic of furthering digital health solutions, which places great emphasis on improved safety, efficacy assessment of endpoints, feasibility and cost-effectiveness measures that can provide real time evaluation [30]. In comparison to the use of repeat radiographic evaluation, surface topography poses no potential for harm, which has been the principal driving motivation in the development of surface topographic methods. Additionally, when compared to radiographs, continually improving surface topographic methods show strong correlation and improvements in software. As a result, ST has become increasingly reliable with easily assessable endpoints to monitor conservative and surgical management [31]. 

Another alternative imaging modality is ultrasound. This also has the advantage of no radiation while also allowing for a potential evaluation of bone quality and metal implants/hardware. Ultrasound, however, requires extensive training to conduct manually and is not clinically viable given its prolonged image processing times. MRI is another alternative that could spare patients radiation, but scan times, cost, hardware, and operating expertise are significant barriers to entry [32]. In this regard, surface topography provides a strong avenue for AIS evaluation. ST allows for real-time data, is generally cheaper than other imaging alternatives, and requires little expertise aside from marker placement as the evaluation is completed by software. ST also shows potential for postural assessment while the patient is walking. 

Limitations with this method come possibly from measurement variations due to patient sway, standing posture, and muscle volume of the back, which still necessitates accompaniment with another imaging modality to assess actual vertebral positioning. Despite its limitations, ST is the most feasible method of evaluation, which provides real-time feedback through many measurable parameters. In future studies, this predictive model will be verified and modified using a new patient pool. In addition, our model may be further strengthened through the addition of factors such as Risser signs, race, or Sanders skeletal maturity, which have been shown to be important in other similar studies. Additionally, the bone mineral density and peak height velocity may contribute to improve the accuracy or specificity of the predictive model. In this study, we do not compare the performance of the final model with other machine learning methods, including radiographic-based or ultrasound-based models. We plan to conduct such a comparative analysis in future studies. 

## 5. Conclusions

Our preliminary finding provides novel bases of algorithm for the use of surface topographic measures in the prediction of future Cobb angle and shows a good digital avenue for the evaluation of spine progression. It also sheds light on developing the artificial intelligence-based assessment of children with AIS using 3D digital metrics of back surface contour. 

## Figures and Tables

**Figure 1 children-11-00159-f001:**
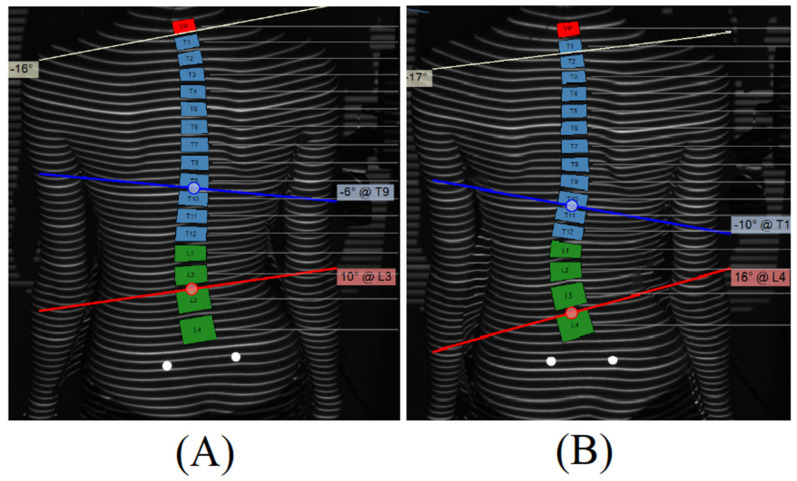
The PA view of surface topography for 14-year-old girl: Scoliotic angle of the left major curve was 16° (T9–L3) at the 1st clinical visit (**A**), while it increased to 26° (T10–L4) at the 2nd clinical visit (**B**).

**Figure 2 children-11-00159-f002:**
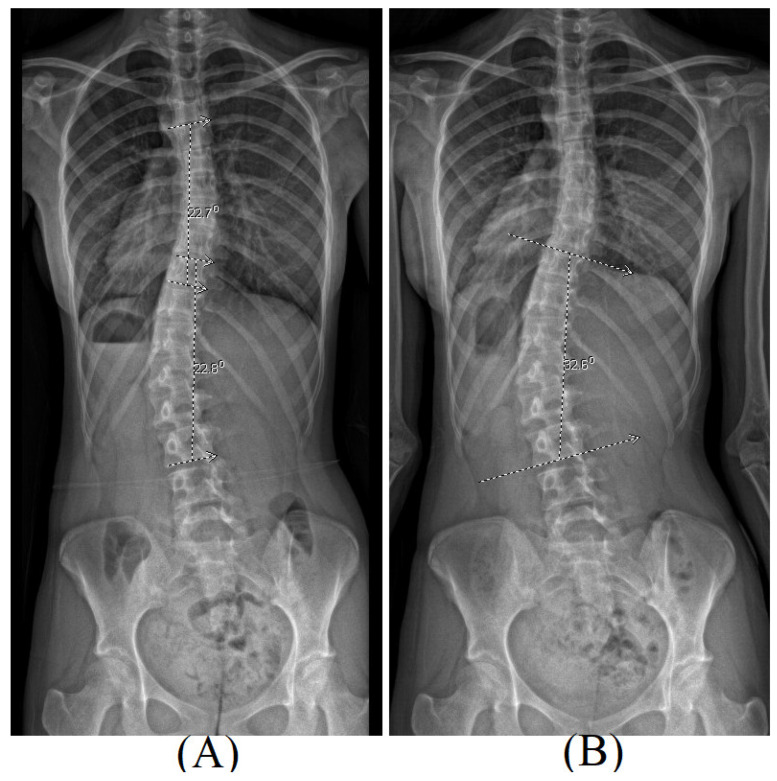
The PA view of radiography for the same patient: Cobb angle of the left major curve was 23° (T10-L3) at the 1st clinical visit (**A**), while it increased to 33° (T10-L3) at the 2nd clinical visit (**B**).

**Figure 3 children-11-00159-f003:**
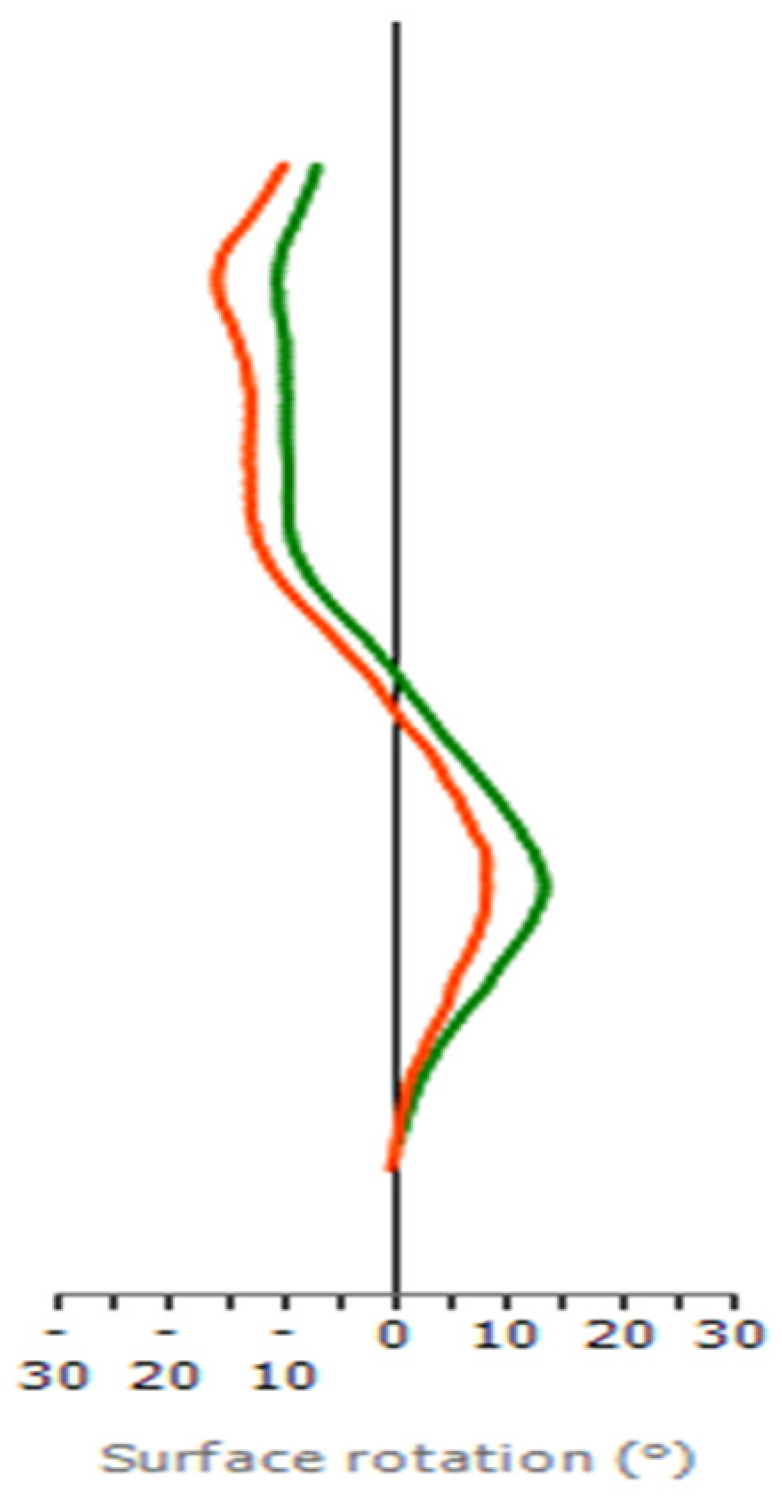
The axial view of surface topography for the same patient: decreases of ASR in the thoracic region (T) but increases of ASR in the thoracolumbar region (TL) from the 1st visit (Orange) to the 2nd visit (Green). Maximal ASR of the TL curve was 8° (L2) at the 1st clinical visit (Orange); however, it increased to 13° (L2) at the 2nd clinical visit (Green). (+) represents a counterclockwise rotation of the vertebral body.

**Table 1 children-11-00159-t001:** Variable in non-radiographic Cobb angle reconstruction (SE, *p* value).

Variable	Estimate	Standard Error	*p*-Value
Intercept	26.243	9.997	0.012
Age	−1.137	0.695	0.109
Gender	−7.689	2.644	0.005
T6	−3.590	1.014	0.001
T7	5.912	1.698	0.002
T8	−2.684	0.869	0.005
Scoliotic angle	0.729	0.093	<0.001

**Table 2 children-11-00159-t002:** The ranges of Y_1_, Y_2_, and Y_3_ for three outcome categories (minimal to maximal value).

Predicting True Outcomes	Y_1_	Y_2_	Y_3_
Progression	(0.034, 1.000)	(<0.001, 0.063)	(<0.001, 0.042)
Stable	(0.008, 0.502)	(0.228, 0.722)	(0.025, 0.664)
Improvement	(<0.001, 0.221)	(0.004, 0.709)	(0.083, 0.996)

Note: Y_1_ (Progression), Y_2_ (Stable), and Y_3_ (Improvement) are probabilities of the progression, stable and improving groups, respectively.

**Table 3 children-11-00159-t003:** Accuracy for stratified predictive model (n, %).

True Group	Classification Based on the Model			Incorrectly Classified n (%)
	Progression n (%)	Stable n (%)	Improvement n (%)	
Progression	6 (75%)	2 (25%)	0 (%)	2 (25%)
Stable	1 (5%)	16 (85%)	2 (10%)	3 (15%)
Improving	0 (0%)	6 (55%)	5 (45%)	6 (55%)

**Table 4 children-11-00159-t004:** Changes of ASR from T8 to L4 measured by surface topography (mean value).

ASR (°)	T8	T9	T10	T11	T12	L1	L2	L3	L4
1st visit	−13	−10	−5	−1	4	7	8	5	2
2nd visit	−9	−7	−3	1	5	10	13	10	4

Note: T—thoracic; L—lumbar; ASR—axial surface rotation.

**Table 5 children-11-00159-t005:** Case report predictive modeling values.

Variable	Y_1_	Y_2_	Y_3_
Probability	0.925	0.073	0.002

## Data Availability

Data can be provided upon reasonable request. The data are not publicly available due to privacy or ethical.

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
