# Peer review of "3D Back Contour Metrics in Predicting Idiopathic Scoliosis Progression: Retrospective Cohort Analysis, Case Series Report and Proof of Concept"

_children, 2024, doi:10.3390/children11020159_

Round 1

Reviewer 1 Report

Comments and Suggestions for Authors

Formal examination:

The topic of the publication is absolutely relevant due to the rarity of works assessing the effectiveness of medical decision support systems in general and in relation to assessing the progression of idiopathic scoliosis in particular.

Design: single-center retrospective cohort analysis based on criteria for AIS progression for machine analysis.

The goal is clearly stated.

Content expertise:

1. Line 92 and 93 may contain an error

2. Main question: To what extent can we speak in general about the primary effectiveness of the proposed criteria only on the basis of one survey method, and also talk about the advantages? It was logical to carry out the work in the design of a comparative analysis of topographic and radiological criteria. This would allow assessment of their reproducibility and specificity in this cohort.

3. There is no indication whether patients received physical therapy methods.

4. In the absence of answers to these questions, I suggest that the authors include these limitations in the title (add “... Case Series Report and Proof of Concept”) and the design of the study. 

General assessment:

Overall, the work is extremely interesting to read and makes a good impression. However, the issues presented prevent its publication in the present form. Please highlight any changes made to the article in color or reflect them in the “review” mode.

I wish the authors success.

Author Response

Dear the Editor and reviewers,

We greatly appreciate your valuable comments!

We have carefully addressed all your concerns and revised them in a point by point response as below. Any change in the manuscript has been highlighted in the yellow color. Additionally, we also ran through a plagiarism checker with a successful pass.

Again, thank you for your consideration of publication!

Sincerely,

Xuecheng

Reviewer 2 Report

Comments and Suggestions for Authors

Identifying predictive elements in the context of patients with juvenile idiopathic scoliosis represents a rapidly expanding field. This article demonstrates the originality of relying on a surface topography technique to predict not the final Cobb angle but stratifying patients into three categories: progression, stability, and improvement. The accuracy of just over 2/3 makes the model comparable to others based on clinical indices and the first x-ray, for example, Eric C. Parent et al., European Spine Journal 2023.

Here are some minor considerations:

- title: specify the study's design

- abstract: "Repeated radiation exposure may be implicated with a modest increase in a child's cancer risk in their formative years." The mSv for each entire spine RX usually ranges from 0.5 to 1.3, and according to the SOSORT guidelines, the frequency depends primarily on age and Risser grade (every 6 to 12 or 18 months). So, it appears more appropriate to state that subjects affected by scoliosis may have an average higher exposure to radiation exposure than unaffected subjects given by the sum of the column's X-ray and the ground exposure.

- introduction: lines 36-37, the paper cited states that there is a small augmentation risk of cancer due to X-ray exposure calculated exposure in patients who finally undergone spine surgery that, in some cases, also done a TC that is known to expose the patient to significant X-ray exposure. The mean episodes for each patient were 8 with a mean exposure of about 21mSv, which means 2.6mSv for each, which seems much higher than the usual range from 0.5 to 1.3mSv for each X-ray of the entire column. Lacking sure the data on this field, it should be better to state as suggested for the abstract or to state that the risk augmentation is, to date, in fact, hypothetically.

- table 2: repeat under the Y1-Y2-Y3 in the table progression, stability and improvement for a major clarity.

- case report: report the distance between the two clinical evaluations. The evaluations with the surface topography happen in the same days as the X-rays? If yes, the difference between estimated and calculated is over 5 degrees, which seems excessive considering that if it is between 16 and 23 degrees, the treatment approach changes according to the SOSORT guidelines.

Author Response

(The authors gave the same response as above.)

Round 2

Reviewer 1 Report

Comments and Suggestions for Authors

The work does not contain a comparative analysis of topographic and radiological criteria. This should be described in the "Limitations" section.

Author Response

We have revised the following in terms of the reviewer 2 and associate editor’s comments and highlighted them in yellow color:

  • To add “no comparative analysis” in the limitation section;
  • To provide more details in the method section;
  • To provide more comprehensive literature review in the introduction section;
  • To provide more discussion for the predictor model as compared to others in the discussion section;
  • Deleted Ref 15 (old) but added Ref 6,11,and 27.